# Assessing progress towards Sustainable Development Goal 3.8.2 and determinants of catastrophic health expenditures in Malaysia

**Muaz Sayuti[1], Surianti Sukeri[2]***

1 Department of Community Medicine and Public Health, Faculty of Medicine and Health Sciences, Universiti Malaysia Sarawak, Kota Samarahan, Sarawak, Malaysia, 2 Department of Community Medicine, School of Medical Sciences, Universiti Sains Malaysia, Kota Bharu, Kelantan, Malaysia

☯ These authors contributed equally to this work.
* surianti@usm.my

**Data Availability Statement:** Data cannot be shared publicly because permission has to be obtained from the Department of Statistics, Malaysia. Enquiries or data requests can be

## Abstract

The Sustainable Development Goal 3.8.2 is an indicator to track a country's progress toward universal health coverage on the financial protection against catastrophic health expenditure (CHE). The purpose of this study is to determine the proportion of households with catastrophic health expenditure, and its associated factors among Malaysian households. A secondary data analysis was performed using the Household Expenditure Survey 2015/2016. The inclusion criterion was Malaysian households with some health spending in the past 12 months before the date of the survey. Catastrophic health expenditure was defined as out-of-pocket health expenditures exceeding 10% of the total household consumption. The study included a total of 13015 households. The proportion of households with CHE in the sample was 2.8%. Female-led households (AdjOR 1.6; CI 1.25, 2.03; p-value <0.001), households in rural areas (AdjOR 1.29; 95% CI 1.04, 1.61; p-value = 0.022), small household size (AdjOR 2.4; 95% CI 1.81, 3.18; p-value <0.001) and heads of household under 60 years old (AdjOR2.34; 95% CI 1.81, 3.18; p-value <0.001) were significantly associated with CHE. Although the proportion of Malaysian households affected by CHE is small, it is increasing in comparison to previous findings. This is concerning because it may jeopardise efforts to achieve universal health coverage by 2030. To ensure financial protection and access to care, a health financing policy that includes safety net measures for households at risk of CHE is required.

## Introduction

In September 2015, the United Nations General Assembly adopted 17 Sustainable Development Goals (SDGs) as a universal call to end poverty, protect the environment, and ensure peace and prosperity for all. The SDG 3 aims to ensure that everyone leads a healthy and happy life, whereas SDG target 3.8 seeks to achieve universal health coverage, which includes financial risk protection, access to quality essential healthcare services, and access to safe, effective, high-quality, and affordable essential medicines and vaccines. The SDG target 3.8

emailed to data@dosm.gov.my (https://www.dosm.gov.my/v1/index.php?r=column/cthree&menu_id=MExaem9UOXM2d0dqSlRBQXdzVEs2UT09).

**Funding:** The author(s) received no specific funding for this work.

**Competing interests:** The authors have declared that no competing interests exist.

includes two indicators: 3.8.1, which measures essential healthcare coverage, and 3.8.2, which measures the proportion of a country's population with catastrophic health expenditures (CHE) [1].

The objective of the study was to assess SDG indicator 3.8.2 for tracking progress towards achieving universal health coverage by 2030. The goal of universal health coverage is to ensure that everyone has access to the health care they require without facing financial hardship. Catastrophic health expenditure is a measure of financial hardship that occurs when out-of-pocket health spending exceeds a certain threshold of a household's ability-to-pay [1]. Ability-to-pay can be expressed as total consumption, total consumption minus food expenditures, total income, or non-food consumption [2]. Out-of-pocket health spending that exceeds 40% of total consumption minus food expenditure and non-food consumption is commonly referred to as catastrophic health expenditure. It can also be defined as out-of-pocket that accounts for more than 5%, 10%, 20%, and 25% of total consumption or income, respectively [3]. The World Bank and World Health Organization used the budget share method of 10% or 25% of total consumption as the CHE definition for SDG3.8.2 [1]. The prevalence of CHE has been increasing, regardless of the threshold or denominator used. More than 800 million people (12% of the world's population) had CHE at 10% of total consumption or income, and 179 million had CHE at 25% of total consumption or income. Globally, the proportion of CHE increased from 9.7% in 2000 to 11.4% in 2005 and 11.7% in 2010. The highest proportion of CHE (14.8%) was found in Latin America, the Caribbean, and Asia (12.8%) while Africa had the fastest increase in the proportion of CHE in its population [1]. Oceania and North America had the lowest proportions of CHE at 3.9% and 4.6%, respectively [3].

Data from a nationally representative household expenditure survey that compiles information on out-of-pocket health spending and other household expenditures can be used to measure the SDG indicator 3.8.2 on the incidence of CHE [1]. It is the recommended source of data to measure the financial burden of healthcare spending on households, which will aid health policymakers in better understanding the effectiveness of various policy instruments and encourage evidence-based policymaking [4]. In Malaysia, according to an unpublished study based on the Household Expenditures Survey 2004/2005, 1.44% of households experienced CHE at a 10% threshold [5]. At the 25% threshold, the WHO Regional Office for the Western Pacific reported 0% CHE [6]. Due to the nature of these reports, there was an incomplete explanation of how the statistics were derived. The low proportion of CHE in Malaysia, on the other hand, could be attributed to the country's health-financing mechanism [3]. As per the Beveridge model, healthcare in Malaysia is primarily funded through general taxation; there is neither social nor national health insurance. With a population of 32.7 million [7], the share of the gross domestic product spent on health in 2019 was 4.3% [8]. The Ministry of Health operates a two-tier healthcare system, with public health care provided at a heavily subsidised rate. Everyone receives free treatment for maternal and child health services, as well as infectious diseases. All treatment-related charges in public healthcare facilities are waived for government employees, disabled people, and the poor. Meanwhile, private health care accounts for 48.9% of total health expenditures and is paid for with non-subsidized out-of-pocket or voluntary private health insurance [9].

Integral to CHE is determining its precursors or associated factors, as these directly identify individuals or households requiring financial assistance. Findings from the literature revealed various factors associated with CHE depending on the type of health expenditures studied (overall health vs. disease specific) and preferred CHE calculation method (definition of ability-to-pay and threshold). A systematic review of 38 studies discovered that the economic status of the household, the frequency of hospitalisation, and the presence of an elderly, disabled, or chronically ill family member were all significant factors associated with household CHE

[10]. Other factors include age, marital status, education, and a higher level of dependence [11], geographic location, household size [12], the absence of local basic medical insurance coverage [13], the types of family structure [14], living arrangements [15], health facility types [16], physical multimorbidity [17], gender of the household head and the quality of life of sick household member [18]. Despite the numerous CHE determinants mentioned in the literature, none have previously been investigated in Malaysia, owing to the low prevalence of CHE and a general lack of interest in the subject.

The outcomes of the study are hoped to fill a knowledge gap on the factors that contribute to CHE, as well as gauge Malaysia's progress toward achieving universal health coverage. The findings of the study will provide evidence-based data that can be used to develop a health financing policy with prepayment mechanisms and safety net measures to protect Malaysian households at risk of CHE.

## Materials and methods

### Source of data

The study design was secondary data analysis. Data was obtained from the Household Expenditure Survey 2015/2016, Department of Statistics Malaysia. This survey gathered information on monthly household consumption, as well as health-related events that occurred within the previous year prior to the survey date. The results of this survey were used by government agencies in the planning, formulation, and implementation of the national development plan. This five-yearly survey was first conducted in 1957 and used probability sampling to represent all Malaysian households. The goal of this survey was to obtain information on households and their spending patterns on a wide range of goods and services. For research purposes, the data is available to interested parties such as economists, academicians, and students [19].

The inclusion criterion was households with some health spending and the exclusion criterion was data with more than 30% missing information. The sample size was calculated using single- and two- proportion formula via the Power and Sample software. The highest sample size required for the study was 9156 households. However, no sampling method was applied; all households whose data fulfilled the study criteria were included in the study.

### Ethical considerations

Ethical approvals were obtained from the Human Research and Ethics Committee, Universiti Sains Malaysia (USM/JEPeM/18100502) and the National Institute of Health, Ministry of Health Malaysia (NMRR-183652-44558).

### Data collection

The data for the Household Expenditures Survey 2015/2016 was provided by Data Mikro, School of Mathematical Sciences, Universiti Sains Malaysia. Prior to data extraction, permission to use the data was obtained. Due to the size of the raw data, it was emailed to the researcher in separate Microsoft Excel files in February 2019. A proforma was used to extract the data. The data acquired include:

- Age, gender, ethnicity, marital status, education, and health insurance ownership of the head of the household

- Household size, income, strata (location)

- All out-of-pocket monthly spending on healthcare and all monthly household expenditures for consumption (personal goods and services including tax, food and house rental fees) and

## Data analysis

Data were screened and imputed in Microsoft Excel spreadsheets. It was then combined and exported to IBM SPSS version 24 software for analysis. Descriptive statistics were used to summarise the socio-demographic characteristics of households. Numerical data were presented as mean (SD) or median (IQR) based on their normality distribution. Categorical data is presented as frequency (%age). The World Bank and World Health Organization budget share method defines CHE as the proportion of the population with large household expenditure exceeding 10% of total household consumption [1].

The associated factors were determined using simple and multiple logistic regression. The dependent variable was CHE (yes/no), whilst independent variables were the head of household's age, gender, marital status, education, health insurance ownership, and household size, residence, monthly income, and income classification based on the Salaries and Wages Survey Report [20]. Variables were chosen using Forward Likelihood selection and Backward Likelihood elimination methods to produce the preliminary final model. The correlation table, standard error, and variance inflation factors were used to test for multicollinearity and interaction between significance variables. All two-way interactions between independent variables were investigated. Subsequently, the fitness of the model was checked using the Hosmer Lemeshow test, classification table and receiver operating characteristic (ROC) curve. The final model of multiple logistic regression was presented as adjusted OR with 95% confident interval, Wald statistic, and corresponding p value.

## Results

This study included a total of 13015 households from the Household Expenditures Survey 2015/2016. Around 10.5% of households were excluded from the study as they did not report any health expenditure in the survey. Table 1 shows the demographic characteristics of households as well as the distribution of monthly out-of-pocket health expenditures. The mean age of the household heads was 47 years old (SD 13.57) and the mean household size was 47 years old (SD 13.57) and the mean household size was 4.14 (SD, 2.05). The majority of household

**Table 1. Demographic characteristics and distribution of monthly out-of-pocket health expenditures and consumption (MYR) (n = 13015).**

| Variable | n (%) | Mean (SD) /Mean (SD) out-of-pocket on health (MYR) | Median (IQR) /Median (IQR) out-of-pocket on health (MYR) |
|---|---|---|---|
| **Age of household head** | | 47.01 (13.57) | 47.00 (20.00) |
| **< 60 years** | 10675 (82.0) | MYR 69.95 (175.28) | MYR 22.20 (57.72) |
| **≥ 60 years** | 2340 (18.0) | MYR 91.66 (174.19) | MYR 30.00 (92.68) |
| **Household size** | | 4.14 (2.05) | 4.00 (2) |
| **Household size classification** | | | |
| **1 to 2** | 2977 (22.9) | MYR 76.40 (167.30) | MYR 24.25 (70.72) |
| **3 to 4** | 5001 (38.4) | MYR 73.34 (165.13) | MYR 24.17 (65.18) |

(*Continued*)

**Table 1.** (Continued)

| Variable | n (%) | Mean (SD) /Mean (SD) out-of-pocket on health (MYR) | Median (IQR) /Median (IQR) out-of-pocket on health (MYR) |
|---|---|---|---|
| **5 and above** | 5037 (38.7) | MYR 72.86 (189.13) | MYR 22.17 (57.43) |
| **Gender of household head** | | | |
| **Male** | 10823 (83.2) | MYR 73.85 (181.65) | MYR 23.33 (61.87) |
| **Female** | 2192 (16.8) | MYR 73.87 (139.66) | MYR 23.78 (67.42) |
| **Head of household ethnicity** | | | |
| **Bumiputra** | 8670 (66.6) | MYR 63.96 (146.03) | MYR 19.99 (52.92) |
| **Chinese** | 2897 (22.3) | MYR 107.78 (218.89) | MYR 42.08 |
| **Indian** | 872 (6.7) | MYR 76.55 (172.78) | (100.15) |
| **Others** | 576 (4.4) | MYR 47.96 (281.61) | MYR 26.69 (60.97) |
| | | | MYR 12.50 (23.77) |
| **Marital status of household head** | | | |
| **Married** | 10072 (77.4) | MYR 75.32 (178.21) | MYR 24.01 (64.72) |
| **Not Married** | 1716 (13.2) | MYR 75.58 (195.59) | MYR 23.00 (59.72) |
| **Divorced/Widowed** | 1227 (9.4) | MYR 59.36 (107.12) | MYR 20.00 (55.05) |
| **Household residence** | | | |
| **Urban** | 9018 (69.3) | MYR 81.36 (191.99) | MYR 27.10 (73.33) |
| **Rural** | 3997 (30.7) | MYR 56.91 (128.32) | MYR 16.66 (41.50) |
| **Education level of household head** | | | |
| **Low Education** | 9870 (75.8) | MYR 63.11 (129.74) | MYR 20.800 |
| **High Education** | 3145 (24.2) | MYR 107.57 (269.87) | (52.67) MYR 34.90 (100.32) |
| **Household income** | | MYR 6300.95 (6010.02) | MYR 4716.75 (4714.50) |
| **Classifications of household income** | | | |
| **Bottom 40%** | 5952 (45.7) | MYR 41.28 (82.33) | MYR 14.53 (33.00) |
| **Middle 40%** | 4910 (37.7) | MYR 77.28 (132.47) | MYR 30.00 (76.05) |
| **Top 20%** | 2153 (16.5) | MYR 156.08 (342.46) | MYR 57.02 (159.95) |
| **Health insurance ownership among head of household** | | | |
| **No health insurance** | 12208 (93.8) | MYR 72.80 (176.15) | MYR 22.95 (61.76) |
| **With health insurance** | 807 (6.2) | MYR 89.79 (160.73) | MYR 32.92 (78.43) |
| | | Mean (SD) | Median (IQR) |
| **Monthly household consumption** | | MYR 3768.73 (2848.49) | MYR 3128.95 (2432.22) |
| **Monthly out-of-pocket health expenditures** | | MYR 73.85 (175.28) | MYR 24.17 (91.67) |

(Conversion rateS MYR1: USD0.24)

heads were under the age of 60 (82%), males (83%), *Bumiputera* (66.6%), married (77.4%), resided in urban areas (69.3%), and had low education levels (75.8%).

The out-of-pocket health expenditures were skewed; means and medians were presented together to allow comparison with national averages when discussing the study findings. According to the findings of this study, the median monthly household income was MYR4716.75 (IQR, 4714.50). Only 6.2% of households had health insurance, and nearly 46% were from the Bottom 40 (B40) income group. According to the findings of this study, the median monthly household expenditure was MYR3128.95 (IQR, 2432.22), and the median monthly out-of-pocket health expenditure was MYR24.12 (IQR, 91.67).

The distribution of median out-of-pocket health expenditures by household size classification, gender, and marital status was generally similar. However, heads of household over the age of 60, Chinese ethnicity, high education, urban households, T20 high income group, and households with health insurance had higher median out-of-pocket health spending than their counterparts. Chinese households spent the most on out-of-pocket health care when compared to other ethnicities, with a median of MYR42.08 (IQR, 100.15). There was a significant disparity in the median of out-of-pocket health expenditures between income groups. The wealthiest (T20) households spent significantly more on health, with a median of MYR57.02 (IQR 159.95) compared to the Bottom 40 (B40) income group, which spent MYR14.53.

## Proportion of households with CHE and its associated factors

The proportion of households with CHE was 2.8% (95% CI 2.5, 3.1). The multiple logistic regression analysis in Table 2 showed that gender, age of the household head, household location, and size were significantly associated with CHE. The model was proved to be fit as the Hosmer-Lemeshow was not significant with p-value 0.82 and the classification table showed 97.2%. The area under the ROC was 68.8% (95% CI 0.659, 0.717; p-value <0.001).

A female head of household had 1.6 times the odds of incurring CHE compared to the male-head of household (95% CI 1.25, 2.03; p-value <0.001) when adjusted for household location, size, and head of household age. Households headed by individuals under the age of

**Table 2. Association between socio-demographic characteristics of households and CHE using multiple logistic regression (n = 13015).**

| Head of household/ household characteristics | β | Adjusted OR (95% CI) | Wald Statistic (df) | p-value |
|---|---|---|---|---|
| Head of household age | | | | |
| ≥ 60 years | | 1 | | |
| < 60 years | 0.850 | 2.34 (1.86, 2.95) | 51.702 (1) | <0.001 |
| Head of household gender | | | | |
| Male | | 1 | | |
| Female | 0.467 | 1.60 (1.25, 2.03) | 14.435 (1) | <0.001 |
| Household residence | | | | |
| Urban | | 1 | | |
| Rural | 0.255 | 1.29 (1.04, 1.61) | 5.234 (1) | 0.022 |
| Household size | | | | |
| 5 and above | | 1 | | |
| 1 to 2 | 0.874 | 2.40 (1.81, 3.18) | 36.737 (1) | <0.001 |
| 3 to 4 | 0.114 | 1.12 (0.84, 1.50) | 0.598 (1) | 0.439 |

Hosmer-Lemeshow test, p-value 0.82.

Classification table 97.2%.

The area under ROC curve 68.8%

60 were 2.34 times more likely to incur CHE compared to those over 60 years old (95% CI 1.86, 2.95; p-value <0.001) when adjusted with gender, location, and household size. A household located in a rural area was 1.29 times more likely to develop CHE compared to a household that lived in an urban area (95% CI 1.04, 1.61; p-value = 0.022) when adjusted for gender, household size, and age of the head of household. A household with 1 to 2 family members was 2.4 times more likely to suffer CHE compared to a household size of 5 and above (95% CI 1.81, 3.18; p-value <0.001) when adjusted for household location, gender, and age of the head of household.

## Discussion

The SDG 3.8.2 indicator on the proportion of households with CHE was 2.8%. In this study, the proportion of households with CHE was higher than the 1.44% [5] and 2.01% [21] estimated in the Household Expenditures Survey 2004/2005 and 1998/1999, respectively. The rising proportion of households with CHE may impede the country's progress toward universal health coverage, jeopardising financial risk protection and access to high-quality essential healthcare services. The increase in the CHE over the years was assumed to be associated with inflation and the rapid expansion of private healthcare in Malaysia, thus increasing the out-of-pocket health expenditures. As theorised by Van Doorslaer et al., [22] out-of-pocket health expenditures influence the trend of CHE. According to the Malaysia National Health Accounts, out-of-pocket health expenditures increased from USD783 million in 1997 to USD5.5 billion in 2019 [8].

However, the proportion of CHE in this study was much lower than the global estimate of 11.7% in the World Bank report from 2010. The low proportion of CHE could be attributed to Malaysia's tax-based health financing system, which provides subsidised healthcare in all public healthcare facilities [23]. Extremely low proportions (0.00%-2.99%) of CHE were observed in countries with nearly identical health financing models such as the United Kingdom, Italy, New Zealand, Denmark, Sweden, Brunei and Saudi Arabia [1]. Our findings reaffirm that increasing the proportion of total health expenditure that is prepaid through taxes leads to a lower rate of CHE [3]. On the other hand, our findings could be an underestimation of the true number of CHE. According to Cylus et al., the budget share method overestimated financial hardship among wealthy households and while underestimating hardship among poor households [24]. This theory, however, was developed among the European population and may not be applicable in the local context. A low prevalence of CHE may also indicate that people are not receiving (or are not paying for) necessary care [3], or that the Household Expenditures Survey respondents were healthy populations who did not require regular medical care. Previous research in Malaysia found that the proportion of CHE was observed to be as high as 47.8% among households affected by colorectal cancer [25], 33.0% for acute gastroenteritis requiring hospitalisation [26] and 16.0% among patients with ischaemic heart diseases [23].

It is critical to discuss out-of-pocket health spending in the Household Expenditures Survey 2015/2016. High out-of-pocket spending was frequently misinterpreted as a negative outcome. According to the Malaysia National Health Accounts, 38% of healthcare expenditure in Malaysia is paid out-of-pocket [8]. However, if one understands the concept of CHE, high out-of-pocket does not always imply catastrophic spending. For example, we discovered that households headed by people under 60 were 2.34 times more likely to suffer from CHE, even though the median out-of-pocket health spending was higher in older and smaller households. This may be because older households require more funds for chronic disease treatment and rehabilitation, which is more common among the elderly. A similar finding was made in a

Malaysian study, which found that those over the age of 50 spent more on health than those under the age of 35 (USD15.98 vs USD9.25). Young Malaysians were discovered to spend a greater proportion of their total spending on vacations, clothing, and entertainment, resulting in a smaller balance of their spending on health [27].

In the current study, Chinese households spent the most on healthcare. In contrast, to the National Health and Morbidity Survey 2015 [28] report, Malay households spent the most on health (USD 125.52; 95% CI, RM0 –USD294.57). According to the same report, Chinese households were the least likely to use government inpatient (43.7%; 95% CI, 29.4–59.0) and outpatient care (52.3%; 95% CI, 43.3–61.1) services, owing primarily to their private health insurance coverage (49.5%; 95% CI, 46.4–52.6).

The current study found that out-of-pocket health expenditures in urban households were higher than in rural households, similar to the findings of the National Health and Morbidity Survey 2015. This could be due to high use of public health services in rural areas or income and spending power disparities between urban and rural populations. This study also found that 44.3% of the rural households were from the low-income group compared with only 36.8% in urban populations. Furthermore, due to high demand, higher concentrations of private healthcare facilities in urban areas resulted in higher out-of-pocket health spending among urban populations. This is supported by the National Health and Morbidity Survey 2015, which found higher utilisation of private healthcare facilities among urban populations (30.4%; 95% CI, 28.2, 32.7) [28, 29].

Highly educated households were found to have more out-of-pocket health expenditures, and the same finding was also reported in the National Health and Morbidity Survey 2015 [28]. The disparity in health spending was due to higher income among most households with a high level of education. Married and unmarried heads of households spent about the same amount on out-of-pocket health care. However, it was discovered to be slightly lower among divorced or widowed heads of households. According to the National Health and Morbidity Survey 2015, high out-of-pocket health spending was recorded among married households [29]. A recent study showed that married households usually have more frequent visits to healthcare facilities compared to unmarried households [30]. The low out-of-pocket health spending among widowed and divorced heads of households may be due to the fact that there are fewer breadwinners in the household, resulting in lower income and spending.

In this study, households with a high income spent more than those with a lower income. This finding is consistent with other research that has found higher out-of-pocket health spending among wealthier households [31–33]. The high spending power of wealthier households is the primary reason that out-of-pocket health expenditures are concentrated among the more affluent population. These high out-of-pocket health expenses will not result in CHE because their total spending was in line with their income. The majority of affluent households use private health facilities, resulting in an increase in the out-of-pocket health spending [31]. Notably, households with health insurance had higher out-of-pocket health expenditures compared to households without health insurance. Further analysis showed that 46.7% of households with no health insurance were from the low-income group, which explained the low out-of-pocket health spending among these households.

Catastrophic health expenditure was more common in female-led households than in male-led households, demonstrating gender disparities in health-care affordability [34]. Despite the fact that out-of-pocket health expenditures were similar in both households, female-led households consumed less than male-led households, which could be due to their lower income. According to the 2016 Salaries and Wages Survey, males earn 7.4% more than females in Malaysia [35], with the median monthly salary for employed Malaysian women

being MYR2254 compared to MYR2345 for men [20]. Nonetheless, a study in Portugal found that male-led households were more likely to incur CHE. This study used a 40% out-of-pocket health expenditures to non-food expenditures as an indicator of CHE, implying that the proportion of CHE was actually higher [36].

Poverty remains mainly a rural challenge. According to the 2016 Household Income Surveys, 17.5% of Malaysians lived in absolute poverty in rural areas, compared to only 4.8% in urban areas [37]. Inadequate access to public services, infrastructure, unemployment, social protection, and economic disparity exacerbate the situation of the rural poor. The median monthly household income in the urban area was MYR5860 compared to MYR3471 in the rural area. This explains why, in this study, rural households were more likely to have CHE, which is consistent with previous findings [14, 16, 34]. The distance to a health facility may also contribute to out-of-pocket health expenses. Even though Malaysia has 1027 health clinics, 87 maternal and child health clinics, 1771 rural clinics, and 286 community clinics, tertiary care is still limited to urban areas. In Malaysia, rural households must travel further to access healthcare than urban families (13.26km vs 9.18km) [29]. However, the cost of travel was not considered when calculating out-of-pocket health expenses in our study. Poverty is more than just a state of being poor; it is a complex process with many facets. Public policy should focus on economic stability, competitive markets, and public investment in physical and social infrastructure, as well as access to land and financing, education, health care, and support services, to eliminate disparities between rural and urban populations [38]

Households with one to two members had a higher risk of CHE than those with five or more members. Similar findings were found in other countries such as Vietnam and Peru [12, 33]. Two assumptions could be used to support this observation. First, family members may be able to better care for one another and encourage a healthy lifestyle in larger households, resulting in lower health-care utilisation. Second, larger families, especially those with working members, can draw on additional resources and share the financial burden during illness and other times of need [33, 36, 39, 40].

The age of the head of household was also found to be linked to the CHE in this study. Individuals under the age of 60 were more likely than those over 60 to develop CHE in their households. In Peru, heads of households between the ages of 18 and 24 were found to be more likely to develop CHE than heads of households between the ages of 45 and 54 [12]. Given that the mean out-of-pocket health expenditures were higher among heads of households over 60, those under 60 were more likely to develop CHE due to the significantly lower total consumption. Furthermore, 75.2% of the low-income group was made up of younger household heads, which may have aided the spread of CHE.

## Study limitations

Although this study used a large national household survey, there were limitations that can be improved in a future study. This study used secondary data obtained from the Household Expenditures Survey 2015/2016 provided by the Department of Statistics. The data in this survey was limited, and it omitted some of the most important characteristics associated with CHE. The survey did not include information on the type of disease or the number of disabled household members. Because this is a household population survey, respondents may be subjected to recall bias, which could affect the accuracy and quality of the data. Despite these disadvantages, a household survey remains the most reliable source of information for determining the CHE among households [1]. Respondents may also fabricate their monthly income and household expenditures as income tax records and purchase receipts were not required to be produced during the survey.

## Conclusions

The proportion of households with CHE in Malaysia was 2.8%. Even though most Malaysians were financially protected from CHE, the proportion of households affected is on the rise. The age and gender of household heads, as well as the size of the household and its location, have all been found to be significant determinants of CHE. They represent Malaysia's most vulnerable families, who require financial assistance to avoid spiralling into debt and a cycle of illness and poverty. In order to make a more accurate analysis of financial risk protection and health coverage, a more detailed study is needed to identify vulnerable populations who may face financial catastrophe as a result of a specific condition. The current health financing structure may be retained but with improvements to prevent abuse. Malaysian public healthcare is struggling to stay afloat due to rising healthcare costs. Malaysians of all income levels use the subsidized healthcare services provided by public hospitals. Hospitals become overcrowded, and health resources become scarce, putting the most vulnerable groups' access to care in jeopardy. Furthermore, households affected by CHE may fall deeper into debt, affecting access to care. A health financing policy is required to prevent abuse and ensure financial assistance is provided to households affected by CHE.

## Acknowledgments

Sincere thanks to Dr. Shamsul Rijal Muhammad Sabri for his assistance in the provision of the Household Expenditures Survey 2015/2016 data.

## Author Contributions

**Conceptualization:** Surianti Sukeri.

**Data curation:** Muaz Sayuti.

**Formal analysis:** Muaz Sayuti.

**Supervision:** Surianti Sukeri.

**Writing – original draft:** Muaz Sayuti.

**Writing – review & editing:** Surianti Sukeri.

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
