## [Decision Letter · Decision Letter 0]

6 Apr 2021

PONE-D-21-06069

Achieving SDG 3.8.2: Financial protection against catastrophic health expenditure in Malaysia

PLOS ONE

Dear Dr. Sukeri,

Thank you for submitting your manuscript to PLOS ONE. After careful consideration, we feel that it has merit but does not fully meet PLOS ONE’s publication criteria as it currently stands. Therefore, we invite you to submit a revised version of the manuscript that addresses the points raised during the review process.

We look forward to receiving your revised manuscript.

Kind regards,

Jahangir Khan

Academic Editor

PLOS ONE

Additional Editor Comments:

Dear authors,

Your manuscript has been reviewed and the comments are accompanied with this decision letter. Submission of a revised version of your manuscript with consideration to the comments from reviewers will be appreciated.

Best regards,

Jahangir Khan

Journal Requirements:

Reviewers' comments:

Reviewer's Responses to Questions

**Comments to the Author**

1. Is the manuscript technically sound, and do the data support the conclusions?

Reviewer #1: No

Reviewer #2: Yes

2. Has the statistical analysis been performed appropriately and rigorously? 

Reviewer #1: No

Reviewer #2: No

3. Have the authors made all data underlying the findings in their manuscript fully available?

Reviewer #1: Yes

Reviewer #2: Yes

4. Is the manuscript presented in an intelligible fashion and written in standard English?

Reviewer #1: Yes

Reviewer #2: Yes

5. Review Comments to the Author

Reviewer #1: Title

Page 1; Line 2 – 3: The current title is not appropriate for this article. The title of this article should read: ‘’Assessing catastrophic health expenditure and its associated factors in Malaysia’’ or ‘’Factors associated with catastrophic health expenditure in Malaysia’’.

Abstract

Background

Page 2: This need to be re-written. Authors should remove SDG 3.8.2.

Method

Page 2: This is confusing. The authors claim they applied the WHO method of calculating CHE yet the authors decided to use a threshold of 10% out-of-pocket health spending from total household expenditures to determine CHE.

Under the WHO methodology, CHE occurs when a household’s total out-of-pocket health payments equal or exceed 40% of household’s capacity to pay or non-subsistence spending. Authors should see: Xu (2005)

Reference: Xu K. (2005). Distribution of Health Payments and Catastrophic Health Expenditures: Methodology. Geneva: World Health Organization.

Results

Page 2: Results are not well presented by the authors. For instance, the statement ‘’the proportion of CHE was 2.8%’’ should read ‘’only 2.8% of households incurred CHE’’.

Conclusion

Page 2: Authors should re-write the conclusion to reflect findings from the study.

Key words:

Page 2: Authors should consider the following keywords: Catastrophic Health Expenditure, Outof-Pocket Payments, Household Survey, Capacity to pay, Universal health coverage, Malaysia.

Background

Overall remark: Authors need to work on the background section. The current presentation of the background section is not acceptable.

Page 3 – 6: The background to the study need to be re-written by the authors as it is not coherent.

Page 4; Line 5 – 7: The authors did not provide a ‘’reference’’ for their claim.

Page 5; Line 17 – 24 and page 6; 1-24 should be moved to the methods section under a subheading titled ‘’Data Source’’.

The authors need to do a literature review of studies conducted on factors associated with CHE and provide a summary of existing literature in the background section.

Method

Page 9; Line 8-9: Authors should see Xu 2005 for an explanation of the WHO approach.

Page 10: Line 1 – 3: What informed the choice of variables? Authors should cite the relevant literature and theories guiding their selection of variables.

Results

Based on the need for a major review of this manuscript, the results of the study is expected to change.

Discussion

The discussion section is most likely going to change since the authors would have more studies (based on extensive literature review) to compare with the findings from this study. � Authors should also include the implications of their findings for policy under the discussion section.

Conclusion

Conclusion of this study is also most likely to change.

Important note

The authors will benefit from an extensive literature review on this topic and the following studies:

1. Myint CY, Pavlova M, Groot W. (2019). Catastrophic health care in Myanmar: policy implications in leading progress towards universal health coverage. International Journal for Equity in Health. 18(11): 118.

2. Salari P, Giorgio LD, Ilinca S, Chuma J. (2019). The catastrophic and impoverishing effects of out-of-pocket healthcare payments in Kenya, 2018. BMJ Global Health 4:e001809. Doi: 10.1136/bmjgh-2019-001809

3. Nundoochan A, Thorabally Y, Monohur S, Hsu J. (2019). Impact of out of pocket payments on financial risk protection indicators in a setting with no user fees: the case of Mauritius. International Journal for Equity in Health 18:63 https://doi.org/10.1186/s12939-019-09595

4. Proano Falconi D, Bernabe E. (2018). Determinants of catastrophic healthcare expenditure in Peru. Int J Health Econ Manag 18: 425-36.

5. Aregbeshola, B. S., & Khan, S. M. (2018). Determinants of catastrophic health expenditure in Nigeria. European Journal Health https://doi.org/10.1007/s10198-017-0899-1

6. Cleopatra I, Eunice K. (2018). Household Catastrophic Health Expenditure: Evidence from Nigeria. Microeconomics and 10.5923/j.m2economics.20180601.01 Macroeconomics. 6(1): 1-8 DOI:

7. Barasa EW, Maina T, Ravishankar N. (2017). Assessing the impoverishing effects, and factors associated with the incidence of catastrophic health care payments in Kenya. International Journal for Equity in Health 16:31. DOI 10.1186/s12939-017-0526-x

8. Amaya-Lara JL. (2016). Catastrophic expenditure due to out-of-pocket health payments and its determinants in Colombian households. International Journal for Equity in Health. 15:182. DOI 10.1186/s12939-016-0472-z

9. Misra, S., Awasthi, S., Singh, J.V., Agarwal, M., Kumar, V. (2015). Assessing the magnitude, distribution and determinants of catastrophic health expenditure in urban Lucknow, North India. Clin. Epidemiol. Glob. Health 3, 10–16.

10. Buigut, S., Ettarh, R., Amendah, D.D. (2015). Catastrophic health expenditure and its determinants in Kenya slum communities. Int. J. Equity Health 14, 46.

11. Brinda, E.M., Andres, R.A., Enemark, U. (2014). Correlates of out-of-pocket and catastrophic health expenditures in Tanzania: results from a national household survey. BMC Int. Health Hum. Rights 14, 5.

12. Li Y, Wu Q, Xu L, Legge D, Hao Y, Gao L, Ning N, Wan G. (2012). Factors affecting catastrophic health expenditure and impoverishment from medical expenses in China: policy implications of universal health insurance. Bull World Health Organ. 90(9):664–71.

Reviewer #2: I thank the authors for this important study on Financial protection against catastrophic health expenditure in Malaysia. The study gives an updated estimate of catastrophic health expenditures using the 2015/2016 HES survey which is necessary for assessing progress towards achieving SDG 3.8.2 and the factors associated with catastrophic health expenditures to identify the vulnerable groups among the Malaysian population who face CHE. These findings are important for planning and designing health financing policies to ensure financial protection among the vulnerable groups in Malaysia. While this is an important study there are areas that needs to be addressed in the different sections of the paper. The following are my review comments that needs to be addressed to help improve the manuscript:

Abstract

• I suggest the first objective should read “to measure the proportion of households with CHE” since in measuring CHE we are trying to assess progress towards UHC

• Sentence number four should be revised to clearly indicate that the authors are using secondary data. Beginning it with “across-sectional study was conducted ….” sounds like the authors conducted the study.

• Sentence number seven needs to be revised to indicate that it is the proportion of sampled households facing CHE and not proportion of CHE

Introduction

• The first Sentence in Paragraph two is not clear it should be revised. One could ask how measuring target 3.8.2 is key to achieving the universal health coverage. Rather it should be clear that measuring target 3.8.3 is key to monitoring progress towards achieving UHC. In the second sentence the last part that reads “until spending on these items is reduced below the level indicated by the poverty line” is unclear it can as well be omitted.

• The last sentence in paragraph three is unnecessary. What is necessary is for the authors to describe the data which they have used in this study in the methods section

• In paragraph seven, sentence beginning with “in Malaysia, the data on OOP….” The information could be placed in the methods section. This will help to free up space to describe the health care systems and financing in Malaysia.

• There is need for part of the information in paragraph seven to be transferred into the methods section and include information on the Health care systems and health systems financing in Malaysia. This will help to give the reader background /context on the health systems and how health care is financed in terms of total health expenditures, how much government, private contributes to total health expenditures.

Methods

• The section on study setting could be described in a paragraph under health systems and health care financing in Malaysia in the introduction section.

• The first sentence under study design and sample should be revised its unclear

• The sentence no sampling method was applied is unclear. This could suggest that the 2015/2016 HES study did not use any sampling methods. I suggest that you read the survey report again and try to describe how sampling for the 2015/2016 HES study was performed and put this information under the data source section

• The section on study design and study sample can come under data source since it is a secondary analysis the study design and the sample can be described under the data source section. Under the section study design and study sample there is a mention of a PS software; it is unclear what PS stands for.

• In the data collection section, the authors may need to provider the recall period for the data on income and all other households ‘expenditures ad may need to indicate whether they are using monthly or annual expenditures

• In the data collection section, the sentence “The data acquired include household’s characteristics “should be revised to indicate that some of the characteristics are for the household head and others are socioeconomic characteristics of the household.

• Under the data analysis section, the authors did describe how income was classified into low, middle and high income groups. Was the classification based on total household consumption or total household consumption per capita or income? This should be clearly described by the authors. In addition, it is not clear whether the authors are using monthly or annual total household expenditures or income. This has to be clarified.

• When defining how CHE was measured it should be made clear that both income and total household expenditure were used since the results section presents the proportion of households that faced CHE based on both income and household expenditures.

Results

• The first sentence in the results section is unnecessary as the sample size has already been given in the methods section. This sentence should be replaced with a sentence that makes reference to the results in table 1 before the results are presented.

• The reported mean age of the household’s head on page 10 does not match with what has been reported in Table1 please check this.

• I suggest reporting the results using Malaysian currency (MYR) and indicating under the table of results that for the year of the survey 1MYR=0.24USD

• The heading for the section on measurement of proportion of households facing CHE should be revised as “Measurement of the proportion of households with CHE”

• As mentioned in point number one you need to make reference to table 2 before presenting the results ;this makes it easier for the reader to follow ie “Table 2 gives of the proportion of households facing CHE….”

• The last sentence under the section measurement of the proportion of households with CHE is unclear it seems there is something missing. It should be revised.

• The second and third sentences under section on Factors associated with CHE are unnecessary and can be omitted as no results on multicollinearity between the variables has been presented.

• The heading for table 3 should be revised to read “Association between socio-demographic characteristics of households and CHE using multiple logistic regression n=13015”

Discussion

• The first sentence could be improved to read “ ……,this study revealed that 2.8% of Malaysian households incurred CHE.

• One of the important/main finding of this study is that the proportion of households that incurred CHE increased in 2015/2016 compared to the estimates in 2004/2008 and 1998/1999; the authors have briefly discussed why the estimate of CHE may have increased, however the implications of this increase has not been discussed. The implications of the increase in CHE needs to be discussed.

• Sentence number 6 in paragraph 1 which begins with “however, the proportion of CHE …..” this should begin as a separate paragraph since the information is comparing CHE estimates in Malaysia and globally while the rest of the information in paragraph 1 is comparing estimate of CHE using 2015/2016 HES to 2004/2008 and 1998/1999 HES

• In paragraph 4 in the second sentence the authors have cited the 2015 National Health Morbidity survey report but no reference has been provided; please provide the reference wherever this report has been cited i.e. in paragraph 5 as well. The use of urbanites in paragraph 4 should be revised and a better word that describe urban residents should be used.

• In paragraph 7 the authors have cited the salaries and wages survey report and no reference has been provided.

• In paragraph 8 the authors provide an important finding of the study in which rural households are more likely to face CHE than urban households. Although the authors have described why this may be so; they have not discussed the policy implications of this finding. The implications of these findings should be discussed.

Conclusion

• In sentence 1 “The SDG 3.8.2” should be omitted and the sentence should be revised as previously suggested.

• There is need to rearrange the sentences to make it easier for the readers to follow for example sentence number 3 in this paragraph should come after sentence one. In addition, the use of despite in sentence number 3 is unnecessary and it can be omitted.

• The last sentence beginning with “A health policy in Malaysia ……..” is very important but it is unclear. It should be revised to indicate the kind of policy and to make it easy for the reader to understand

Minor comments

• There are a number of grammatical errors in the discussion; the paper could benefit a lot from English editorial services. This will make the discuss section easy to follow.

• Authors should also check the use tenses in the paper.

• When resubmitting please use line numbers for easy reference when reviewing the manuscript.

6. PLOS authors have the option to publish the peer review history of their article (what does this mean?). If published, this will include your full peer review and any attached files.

Reviewer #1: **Yes: **Bolaji Samson Aregbeshola

Reviewer #2: No

---

## [Author Response · Author response to Decision Letter 0]

19 Sep 2021

Thank you for all the constructive comments provided by the reviewer. The Introduction had been fully re-written to comply with the recommendations by reviewers. English editing and grammar check were performed by Quill Bot.

Thank you.

---

## [Decision Letter · Decision Letter 1]

2 Feb 2022

PONE-D-21-06069R1Assessing progress towards Sustainable Development Goals 3.8.2 and determinants of catastrophic health expenditures in MalaysiaPLOS ONE

Dear Dr. Sukeri,

Thank you for submitting your manuscript to PLOS ONE. After careful consideration, we feel that it has merit but does not fully meet PLOS ONE’s publication criteria as it currently stands. Therefore, we invite you to submit a revised version of the manuscript that addresses the points raised during the review process.

We look forward to receiving your revised manuscript.

Kind regards,

Tariq Jamal Siddiqi

Academic Editor

PLOS ONE

Journal Requirements:

Reviewers' comments:

Reviewer's Responses to Questions

**Comments to the Author**

1. If the authors have adequately addressed your comments raised in a previous round of review and you feel that this manuscript is now acceptable for publication, you may indicate that here to bypass the “Comments to the Author” section, enter your conflict of interest statement in the “Confidential to Editor” section, and submit your "Accept" recommendation.

Reviewer #2: All comments have been addressed

2. Is the manuscript technically sound, and do the data support the conclusions?

Reviewer #2: Yes

3. Has the statistical analysis been performed appropriately and rigorously? 

Reviewer #2: Yes

4. Have the authors made all data underlying the findings in their manuscript fully available?

Reviewer #2: Yes

5. Is the manuscript presented in an intelligible fashion and written in standard English?

Reviewer #2: Yes

6. Review Comments to the Author

Reviewer #2: The authors have addressed all the comments accordingly. The manuscript is ready for publication. However, i suggest that the authors should go through the manuscript again to check for grammatical errors. For example please check and revise line191,lines 62-62,line 64 and lines 65-66

7. PLOS authors have the option to publish the peer review history of their article (what does this mean?). If published, this will include your full peer review and any attached files.

Reviewer #2: No

---

## [Author Response · Author response to Decision Letter 1]

3 Feb 2022

Response to reviewer 

There was only one comment as listed below.

Reviewer #2: The authors have addressed all the comments accordingly. The manuscript is ready for publication. However, i suggest that the authors should go through the manuscript again to check for grammatical errors. For example please check and revise line191,lines 62-62,line 64 and lines 65-66

Response: The whole manuscript has been English edited and proofread using Quillbot and Grammarly. All grammar errors have been corrected and sentences were restructured.

---

## [Editor Report · Decision Letter 2]

11 Feb 2022

Assessing progress towards Sustainable Development Goals 3.8.2 and determinants of catastrophic health expenditures in Malaysia

PONE-D-21-06069R2

Dear Dr. Sukeri,

We’re pleased to inform you that your manuscript has been judged scientifically suitable for publication and will be formally accepted for publication once it meets all outstanding technical requirements.

Kind regards,

Tariq Jamal Siddiqi

Academic Editor

PLOS ONE
---

## [Editor Report · Acceptance letter]

16 Feb 2022

PONE-D-21-06069R2 

Assessing progress towards Sustainable Development Goal 3.8.2 and determinants of catastrophic health expenditures in Malaysia 

Dear Dr. Sukeri:

I'm pleased to inform you that your manuscript has been deemed suitable for publication in PLOS ONE. Congratulations! Your manuscript is now with our production department. 

Kind regards, 

on behalf of

Dr. Tariq Jamal Siddiqi 

Academic Editor

PLOS ONE